# Preoperative diagnoses and identification rates of unexpected gallbladder cancer

**Kenji Fujiwara** ¤*, **Toshihiro Masatsugu, Atsushi Abe, Tatsuya Hirano, Masayuki Sada**

Department of Surgery, Sada Hospital, Fukuoka, Japan

¤ Current address: Collaborative researcher in the Department of Surgery and Oncology, Graduate School of Medical Sciences, Kyushu University, Fukuoka, Japan
* kengdom@surg1.med.kyushu-u.ac.jp

## Abstract

### Background

Unexpected gallbladder cancer (UGBC) is sometimes found in the resected gallbladder of patients during or after surgery. Some reports have indicated UGBC identification rates for all gallbladder surgeries, but scarce data are available for the UGBC identification rates for specific gallbladder diseases. The present study was performed to clarify UGBC rates and the factors suspicious for UGBC categorized according to preoperative diagnoses, in patients undergoing laparoscopic cholecystectomy (LSC).

### Methods

We recorded data for all LSC surgeries performed in the Department of Surgery, Sada Hospital, Japan since 1991, and analyzed the 28-year data. We used the chi-square test and Kaplan–Meier analysis for this retrospective case–control study.

### Results

The UGBC identification rate was 0.69% (63/9186 patients). The UGBC identification rates categorized according to the preoperative diagnoses were 1.3% (13/969) for acute cholecystitis, 2.4% (16/655) for benign tumor, 2.0% (28/1383) for chronic cholecystitis or cholecystitis, and 0.054% (3/5585) for cholecystolithiasis. The percentage of older patients ($\geq$ 60 years) was significantly higher in UGBCs compared with cases finally diagnosed as benign in each group categorized according to the preoperative diagnoses (p$\leq$0.0014), except for cholecystolithiasis. In cases pre-diagnosed as benign tumor, UGBCs were associated with higher rates of thickened gallbladder wall compared with benign tumor (69.2% vs. 27.9%, respectively; p = 0.0011). UGBCs pre-diagnosed as acute cholecystitis had higher T2–T4 rates (100% vs. 64.3%, respectively; p<0.05) and lower survival rates (p = 0.0149) than UGBCs pre-diagnosed with chronic cholecystitis.

### Conclusions

UGBC identification rates depend on the preoperative diagnosis and range from 0.054% to 2.4%. Older age ($\geq$ 60 years) could be related to UGBC, and a pre-diagnosis of acute

**Data Availability Statement:** Original full data cannot be shared publicly because of containing detailed surgical information like precise surgical date. This information may indicate the patients' information. Data after being anonymized are

available from the Sada Hospital Institutional Review Board (contact via e-mail; info@sada.or.jp) for researchers who meet the criteria for access to confidential data.

**Funding:** The authors received no specific funding for this work.

**Competing interests:** The authors have declared that no competing interests exist.

cholecystitis might indicate more advanced cancer compared with a pre-diagnosis of chronic cholecystitis.

## Introduction

Gallbladder cancer is sometimes detected in the resected gallbladder. High-quality imaging techniques such as computed tomography (CT) and magnetic resonance imaging (MRI) can be used to verify suspected cases of gallbladder cancer before surgery [1]. In such cases, patients and surgeons can discuss the best way to cure the patient before surgery and adequately prepare for surgery. However, we still experience cases involving sudden identification of gallbladder cancer during or after surgical treatment, termed unexpected gallbladder cancer (UGBC) [2, 3].

Providing patients with detailed UGBC identification rates preoperatively may decrease postoperative confusion regarding UGBC and patients' perceived risk of malignancy. Some studies reported UGBC identification rates, but most reported data only for all gallbladder surgeries, and scarce data are available for the UGBC identification rate for each gallbladder disease [4–9]. If the identification rates differ among preoperative diagnoses, and if we clarify the rates, surgeons will be able to provide more concrete data to patients.

Löhe et al. [10] reported that only 50% of gallbladder cancers are recognized preoperatively. There could be several reasons for the difficulty recognizing cancers preoperatively. For example, the thickened tumor wall is sometimes difficult to distinguish from wall thickening owing to inflammation in cholecystitis [11]. Additionally, early-stage cancers and/or flat-type cancers are difficult to diagnose [11, 12]. Several reports showed the risk factors for malignancy of gallbladder lesions [2, 3]. In this study, we evaluated the efficacy of using the reported risk factors to predict the UGBC and to determine the factors suspicious for cancer, especially when categorizing the risk factors according to preoperative diagnoses. In addition, we compared the pathological findings and the prognosis of gallbladder cancer cases with preoperative diagnoses. Identifying differences in gallbladder cancer progression according to preoperative diagnoses might provide clues to determine poor prognostic factors.

## Methods

### Patients' characteristics

In total, 9200 LSC procedures (including conversion to laparotomy) were performed at Sada Hospital, Fukuoka, Japan during the 28 years from 31 January 1991 to 22 July 2019. Patients' demographic details with final diagnoses are provided in S1 Table. Patients' ages ranged from 3 to 97 years (median, 55 years). Patients comprised 4909 women (53.4%) and 4291 men (46.6%), and female patients tended to be older than the male patients (median, 56 years vs. 54 years, respectively; S1 Fig). The final diagnoses of the 9200 patients undergoing LSC are summarized in Table 1.

### Database management

All surgical data for LSCs are maintained in our hospital. These data include patients' diagnoses, sex, age, symptoms, imaging findings, adhesion to surrounding tissues, and presence/absence of stones. Wall thickness was determined postoperatively by evaluating the surgical specimen. For cases pre-diagnosed definitively as gallbladder cancer, laparotomy was

**Table 1. Breakdown of final diagnoses among patients who underwent LSC.**

| Final diagnosis | N (%) | Age range (median), (years) | Percentage of women |
|---|---|---|---|
| Cholecystolithiasis and choledocholithiasis | 5582 (60.7%) | 3–97 (55) | 58.1% |
| Chronic cholecystitis /cholecystitis | 1355 (14.7%) | 20–97 (56) | 49.3% |
| Acute cholecystitis | 956 (10.4%) | 15–95 (59) | 37.6% |
| Benign tumor | 639 (6.9%) | 7–91 (48) | 43.5% |
| Adenomyomatosis | 359 (3.9%) | 18–89 (47) | 51.8% |
| Gallbladder cancer | 77 (0.84%) | 46–87 (69) | 53.2% |
| Others | 9 (0.098%) | 25–65 (40) | 77.8% |
| Unknown | 223 (2.4%) | 16–87 (56) | 57.4% |
| Total | 9200 | 3–97 (55) | 53.4% |

LSC, laparoscopic cholecystectomy.

performed, and data for these patients were excluded from our LSC database. Tumor–node–metastasis (TNM) stage was classified according to the 8th edition of the American Joint Committee on Cancer (AJCC) staging of gallbladder cancer. Diagnostic terms were determined at the end of each surgery by each operating surgeon according to the patient's clinical history, blood test data, imaging data (ultrasonography, CT, and/or MRI), intraoperative intraabdominal findings, and postoperative macroscopic observations of the resected specimen. When the results of the pathological data showed that the specimen was cancer, we changed the diagnostic terms to include additional information. In our facility, the term "acute cholecystitis" was used for cases with acute progression of the disease with abdominal pain, systemic signs of inflammation (fever, elevated white blood cell count, or increased C-reactive protein concentration), and imaging findings of inflammation, such as a thickened edematous gallbladder wall. The term "chronic cholecystitis" was used for cases with findings of a thickened wall and atrophic or fibrotic gallbladder and without systemic signs of inflammation. The term "cholecystitis" was used for patients whose gallbladders showed evidence of inflammation, such as an edematous wall or adhesion to surrounding tissues but without an increase in systemic signs of inflammation. The term "cholecystolithiasis" referred to patients with gallbladder stones with or without symptoms but without evidence of acute gallbladder inflammation and without an increase in systemic signs of inflammation.

## Analyzing the database for the UGBC study

For this analysis, we categorized patients' data into seven diseases: 1) cholecystolithiasis and choledocholithiasis, 2) chronic cholecystitis and cholecystitis, 3) acute cholecystitis, 4) benign tumor (including gallbladder polyp), 5) adenomyomatosis, 6) gallbladder cancer, and 7) others (biliary dyskinesia and congenital biliary dilatation). Many patients with choledocholithiasis had gallbladder stones, so we categorized both "cholecystolithiasis" and "choledocholithiasis" as one category. The difference between "cholecystitis" and "chronic cholecystitis" was vague when we evaluated the medical records; therefore, we categorized these diagnoses into one category. "Benign tumor" meant gallbladder protruded lesions, including gallbladder pseudo-polyps such as cholesterol polyp and also true polyps such as adenomas, pathologically diagnosed as benign. We categorized "adenomyomatosis" separately from "benign tumor" because typical adenomyomatosis lesions are distinguishable from gallbladder polyps by imaging [13]. We found 1239 cases (13.5% in 9200 cases) with two or three diagnostic terms simultaneously, and we summarized these in S2 Table. We prioritized the main diagnostic terms, such as cholecystitis, when patients had two diagnostic terms, such as gallbladder polyp, coincidentally found

in the gallbladder with cholecystitis. When patients had multiple diagnostic terms, including choledocholithiasis, we prioritized other diagnostic terms because gallbladder lesions are usually related to the risk of malignancy more than with choledocholithiasis. In cases with concurrent cholecystolithiasis and gallbladder polyp, we chose gallbladder polyp (as a benign tumor) to categorize the disease according to the preoperative diagnosis because gallbladder polyp may carry a higher risk of malignancy. Some patients' records had no diagnostic terms, so we defined cases with gallbladder stones as "cholecystolithiasis".

To analyze the data for gallbladder cancers according to the preoperative diagnosis, we categorized the data based on patients' symptoms, blood test results, imaging findings, and preoperative diagnoses. We used the 2018 Tokyo guidelines criteria for acute cholecystitis to select patients with UGBC who were thought to have acute cholecystitis preoperatively [14]. In the database, information describing preoperative/intraoperative findings was missing for some patients, so some tables in this manuscript show different total numbers. Survival was determined from the date of surgery to the date of death or to the date of the last follow-up.

### Ethics statement

The Sada Hospital Institutional Review Board (IRB) reviews all studies in our hospital. The IRB approved the use of the database for research purposes and waived the requirement for informed consent (IRB number: S190726-1). We fully anonymized all data before accessing the data.

### Statistical analysis

Patients' characteristics and pathological findings were analyzed using the Chi-square test. Survival data for the Kaplan–Meir analysis was performed using the log-rank test. Statistical analysis and graphic presentations were performed using GraphPad Prism Version 8.3.1 (GraphPad Software, San Diego, CA, USA). A p-value of $<0.05$ was considered statistically significant.

## Results

### UGBC identification rates categorized according to preoperative diagnoses

To analyze the data for gallbladder cancers according to the preoperative diagnostic terms, we categorized 77 gallbladder cancer cases according to the preoperative diagnoses and calculated the UGBC identification rates for each preoperative diagnosis (Table 2 and S3 Table). Of 9186 cases, after excluding 14 gallbladder cancer patients suspected preoperatively of having malignancy, the number of UGBC patients was 63, and the UGBC identification rate was 0.69%. The UGBC identification rate in patients with benign tumor was highest (2.4%), followed by

**Table 2. Identification rates of unexpected gallbladder cancer categorized according to the preoperative diagnosis.**

| Preoperative diagnosis | Percentages (UGBC numbers/total counts) |
|---|---|
| Cholecystolithiasis and choledocholithiasis | 0.054% (3/5585) |
| Chronic cholecystitis/ cholecystitis | 2.0% (28/1383) |
| Acute cholecystitis | 1.3% (13/969) |
| Benign tumor | 2.4% (16/655) |
| Adenomyomatosis | 0.83% (3/362) |
| Total | 0.69% (63/9186) |

chronic cholecystitis or cholecystitis (2.0%), and acute cholecystitis (1.3%). The UGBC identification rate in cases with cholecystolithiasis was lowest (0.054%).

## Factors indicating possible UGBC

We analyzed pre/postoperative findings in each patient categorized according to the final diagnosis (S4 Table). Preoperative data included age, sex, and gallbladder imaging on drip infusion cholangiography with CT (DIC-CT), and gallbladder wall thickness was confirmed after surgery by evaluating the surgical specimen. The data showed that 1) the percentage of older adult patients ($\geq$ 60 years) was significantly higher in gallbladder cancer patients compared with the other diseases (80.5% vs. 11.1%–47.8%, respectively; p<0.001). 2) DIC-CT showed higher rates of gallbladder-negative contrast in patients with acute cholecystitis (83.3%, p<0.001) and gallbladder cancer (48.7%) compared with the other diseases (0%–36%). 3) Patients with acute cholecystitis showed the highest detection rate of a thickened gallbladder wall (95.5%) compared with the other diseases (27.9%–87.5%; p<0.001).

To compare pre/postoperative findings between patients finally diagnosed as benign and the UGBC patients, we summarized the factors in gallbladder-cancer patients by categorizing according to the preoperative diagnoses (S5 Table). Comparing the data in S4 and S5 Tables showed that the percentage of older patients ($\geq$ 60 years) was significantly higher in UGBCs (68.7%–100%) than in cases finally diagnosed as benign (21.2%–47.8%) in each preoperative diagnosis group (p$\leq$ 0.0014) except for UGBC patients pre-diagnosed with cholecystolithiasis (Table 3). The detection rate of a thickened gallbladder wall in the UGBC patients pre-diagnosed with benign tumor was significantly higher compared with the rate in the patients finally diagnosed as benign (69.2% vs. 27.9%, respectively; p = 0.0011). Regarding sex, the data showed no significant difference (S6 Table). The rate of gallbladder negative-contrast on DIC-CT imaging was significantly higher only in the UGBC patients pre-diagnosed with benign tumor compared with the patients finally diagnosed as benign (28.6% vs. 4.8%, respectively; p = 0.006).

## Analysis of pathological findings and prognosis of UGBC

We analyzed the pathological findings of 77 gallbladder cancer cases, including 14 with preoperatively suspected malignancy and 63 UGBCs (Table 4). Sixty-two cases (80.5% of the 77 cases) were higher than stage T2, and 14 of 41 (34.1%) cases with information describing

**Table 3. Comparison of pre/postoperative findings between the patients finally diagnosed as having unexpected gallbladder cancer (UGBC) with the patients finally diagnosed as having benign disease.**

| Preoperative diagnosis | Final diagnosis | Age ($\geq$60 years) | p-value | Thickened wall | p-value |
|---|---|---|---|---|---|
| Cholecystolithiasis and choledocholithiasis | UGBC | 2/3 (66.7%)* | 0.30 | 3/3 (100%) | 0.15 |
| | Benign | 2090/5582 (37.4%) | | 2979/5070 (58.8%) | |
| Chronic cholecystitis and cholecystitis | UGBC | 23/28 (82.1%) | <0.001 | 16/25 (64.0%) | 0.30 |
| | Benign | 555/1355 (41.0%) | | 986/1345 (73.3%) | |
| Acute cholecystitis | UGBC | 12/13 (92.3%) | 0.0014 | 12/12 (100%) | 0.45 |
| | Benign | 457/956 (47.8%) | | 864/905 (95.5%) | |
| Benign tumor | UGBC | 11/16 (68.7%) | <0.001 | 9/13 (69.2%) | 0.0011 |
| | Benign | 141/639 (22.1%) | | 163/584 (27.9%) | |
| Adenomyomatosis | UGBC | 3/3 (100%) | <0.001 | 3/3 (100%) | 0.51 |
| | Benign | 76/359 (21.2%) | | 293/335 (87.5%) | |

*Numbers indicate the number of positive cases/total cases (%).

**Table 4. Pathological findings of gallbladder cancer cases categorized according to the preoperative diagnoses.**

| Preoperative diagnosis | T factor stage | | N factor stage | | |
|---|---|---|---|---|---|
| | Tis–T1 | T2–4 | N0 | N1/N2 | NX |
| Gallbladder cancer suspected (N = 14) | 1 (7.1%) | 13 (92.9%) | 5 (35.7%) | 4 (28.6%) | 5 (35.7%) |
| Chronic cholecystitis and cholecystitis (N = 28) | 10 (35.7%) | 18 (64.3%) | 13 (46.4%) | 2 (7.1%) | 13 (46.4%) |
| Acute cholecystitis (N = 13) | 0 | 13 (100%) | 1 (7.7%) | 5 (38.5%) | 7 (53.8%) |
| Benign tumor (N = 16) | 4 (25.0%) | 12 (75.0%) | 6 (37.5%) | 2 (12.5%) | 8 (50.0%) |
| Cholecystolithiasis and choledocholithiasis (N = 3) | 0 | 3 (100%) | 1 (33.3%) | 0 | 2 (66.7%) |
| Adenomyomatosis (N = 3) | 0 | 3 (100%) | 1 (33.3%) | 1 (33.3%) | 1 (33.3%) |
| Total (N = 77) | 15 (19.5%) | 62 (80.5%) | 27 (35.1%) | 14 (18.2%) | 36 (46.8%) |

lymph nodes had lymph node metastases (N1 or N2). Patients pre-diagnosed as acute cholecystitis and gallbladder-cancer suspected patients showed significantly higher rates of stage T2–T4 cancer (100%, p = 0.013 and 92.9%, p = 0.047, respectively) and higher percentages of lymph node metastases (38.5% and 28.6%, respectively) compared with UGBC patients pre-diagnosed as chronic cholecystitis/cholecystitis (T2–T4 cases: 64.3% and lymph node metastases: 7.1%). We compared the survival curves of the cancer patients categorized according to the preoperative diagnoses (Fig 1). Cases pre-diagnosed as acute cholecystitis and cancer-suspected cases showed significantly lower survival rates than cases pre-diagnosed as chronic cholecystitis/cholecystitis (p = 0.015, p = 0.008, respectively).

## Discussion

In this study, the UGBC identification rate during or after surgery in patients undergoing LSC was 0.69%. The term UGBC also indicates occult gallbladder cancer detected pathologically after surgery [5, 15–17]; therefore, we analyzed the incidental pathological UGBC detection rate in our facility; the rate was 0.43% (S7 Table). In previous reports, the UGBC identification rate during or after surgery was 1.0%–2.1% [18, 19], and the incidental pathological UGBC

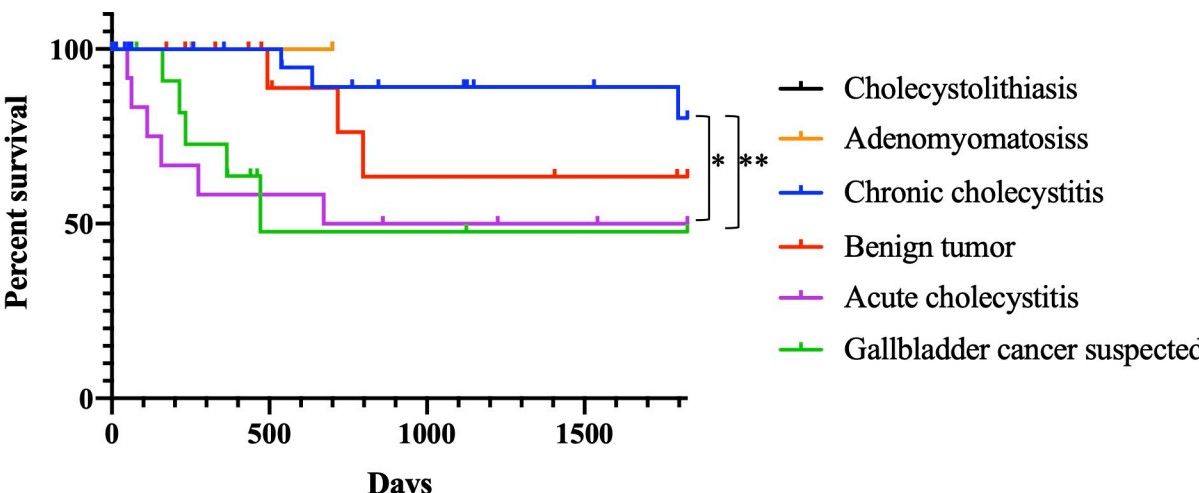

**Fig 1. Kaplan–Meier survival curves in unexpected gallbladder cancer (UGBC) cases categorized according to the preoperative diagnoses.** *p<0.05, **p<0.01.

detection rate was 0.25%–1.0% [4, 15, 16, 20]. Our UGBC identification rate was low compared with the rates in these studies. We suspect that the type of gallbladder disease in each facility could affect this rate. In our LSC cases, cholecystolithiasis was dominant (60.7%), and we found that UGBC was rarely found in cholecystolithiasis cases (0.054%). Kim et al. reported a percentage of acute cholecystitis in their LSC cases of 43.3%, which was much higher than ours (10.3%), and their incidental pathological UGBC detection rate was 1.0% [21]. Differences in gallbladder diseases could have affected the differences in UGBC rates between the studies.

After categorizing UGBC detection rates according to the preoperative diagnoses, the data showed that older age ($\geq$ 60 years) was a possible factor indicating UGBC for most diagnoses. Our data also showed that UGBCs diagnosed preoperatively as benign tumors had thickened walls more frequently than cases finally diagnosed as benign disease. Although we examined thickened walls postoperatively, other studies reported that a preoperative finding of thickened walls could also indicate UGBC [21, 22]. However, a thickened wall may not be a useful UGBC risk factor for other UGBC cases, such as UGBCs in acute cholecystitis. Resected specimens in acute cholecystitis usually have thickened walls (95.5% in our data).

Identifying the current UGBC identification rates (especially by categorizing rates according to the preoperative diagnoses) was our first purpose in this study. Our UGBC identification rate in acute cholecystitis cases was 1.3%, and the rate was higher than the 1.0% reported by Thorbjarnarson in 1960 [23]. Kim et al. also reported a high percentage of UGBC in acute cholecystitis (1.6%) in 2010 [15]. We do not have a clear answer regarding the increased identification rate of UGBC in the patients diagnosed with acute cholecystitis during the past half-century. We believe that even high-definition imaging cannot sufficiently distinguish the thickened walls of gallbladder cancers from those of acute cholecystitis preoperatively, but further evaluation is needed to determine the reason for the increased rate [11].

We acknowledge that this study has limitations regarding the accuracy of the diagnostic terms. We used the diagnostic terms in the records reported by each surgeon, but the recorded diagnostic terms were not reassessed by other surgeons. Revalidation of all patients' data was difficult because of the loss of records owing to storage expiration dates during the study period. A second limitation was the insufficient numbers of UGBC cases for evaluating the risk factors. We aimed to analyze data from as large a base as possible, so we decided to use the entire LSC database despite the difficulty confirming patients' diagnoses. We kept all records for cancer cases during the 28-year period, and we are confident in the preoperative diagnoses for these cases. The entire database has been managed and supervised by one specific surgeon continuously for 28 years; therefore, we believe that the criteria remained the same for the diagnoses.

In conclusion, the UGBC identification rates among the preoperative diagnoses of gallbladder disease varied widely from 0.054% to 2.4%. Older age was a factor indicating UGBC for most of the preoperative diagnoses. In addition, a pre-diagnosis of acute cholecystitis might indicate more advanced cancer compared with a pre-diagnosis of chronic cholecystitis. We hope that our data will support the sharing of information between patients and surgeons.

## Supporting information

**S1 Fig. Age distribution curve of the patients who underwent LSC.** All patients (bar graphs), male patients (dotted black line), and female patients (solid red line).
(TIFF)

**S1 Table. Patients' demographics.**
(DOCX)

**S2 Table. Numbers of patients with multiple diagnostic terms.**
(DOCX)

**S3 Table. Preoperative diagnoses of the 77 patients with gallbladder cancer.**
(DOCX)

**S4 Table. Pre/postoperative findings for the patients undergoing laparoscopic cholecystectomy (LSC), categorized according to the final diagnoses.**
(DOCX)

**S5 Table. Pre/postoperative findings for gallbladder cancer patients categorized according to the preoperative diagnoses.**
(DOCX)

**S6 Table. Comparison of pre/postoperative findings between the patients finally diagnosed as having unexpected gallbladder cancer (UGBC) with the patients finally diagnosed as having benign disease (sex and gallbladder imaging on drip infusion cholangiography with computed tomography (DIC-CT).**
(DOCX)

**S7 Table. Identification rates of incidental pathologically-detected gallbladder cancer cases categorized according to the preoperative diagnoses.**
(DOCX)

## Acknowledgments

We thank Ms. Kazuko Kido, who worked in data management at Sada Hospital for all 28 years of the study period. We also thank all medical staff, nurses, and surgeons who were involved in performing the LSC surgeries and developing the database throughout the 28-year study period. We thank Jane Charbonneau, DVM, from Edanz Group (https://en-author-services.edanzgroup.com/ac) for editing a draft of this manuscript.

## Author Contributions

**Conceptualization:** Kenji Fujiwara.

**Data curation:** Kenji Fujiwara, Toshihiro Masatsugu, Atsushi Abe, Tatsuya Hirano, Masayuki Sada.

**Formal analysis:** Kenji Fujiwara.

**Methodology:** Kenji Fujiwara.

**Project administration:** Kenji Fujiwara.

**Supervision:** Masayuki Sada.

**Visualization:** Kenji Fujiwara.

**Writing – original draft:** Kenji Fujiwara.

**Writing – review & editing:** Toshihiro Masatsugu, Atsushi Abe, Tatsuya Hirano, Masayuki Sada.

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
