## [Decision Letter · Decision Letter 0]

1 Jul 2020

PONE-D-20-15344

Preoperative Diagnoses of and Identification Rates of Unexpected Gallbladder Cancer

PLOS ONE

Dear Dr. Fujiwara,

Thank you for submitting your manuscript to PLOS ONE. After careful consideration, we feel that it has merit but does not fully meet PLOS ONE’s publication criteria as it currently stands. Therefore, we invite you to submit a revised version of the manuscript that addresses the points raised during the review process.

We look forward to receiving your revised manuscript.

Kind regards,

Ker-Kan Tan

Academic Editor

PLOS ONE

Additional Editor Comments:

Dear Authors,

Many thanks for submitting this manuscript to PLOS ONE

Perhaps, it would be good to try to shorten the initial part of the paper from Introduction to Results

For the discussion section, I think it would be best for the authors to highlight what are the implications of the findings of the paper and the subsequent research questions that should be answered going ahead.

And do consider the various comments from the reviewers

'Our institutional review board approved the use of this database for research purposes and waived the requirement for informed consent.'   

(a) Please amend your current ethics statement to include the full name of the ethics committee/institutional review board(s) that approved your specific study.

(b) Once you have amended this/these statement(s) in the Methods section of the manuscript, please add the same text to the “Ethics Statement” field of the submission form (via “Edit Submission”).

For additional information about PLOS ONE ethical requirements for human subjects research, please refer to " ext-link-type="uri" xlink:type="simple">http://journals.plos.org/plosone/s/submission-guidelines#loc-human-subjects-research."

3. In your ethics statement in the manuscript and in the online submission form, please provide additional information about the patient records used in your retrospective study.

Specifically, please ensure that you have discussed whether all data were fully anonymized before you accessed them.

4. To comply with PLOS ONE submission guidelines, in your Methods section, please provide additional information regarding your statistical analyses.

For more information on PLOS ONE's expectations for statistical reporting, please see https://journals.plos.org/plosone/s/submission-guidelines.#loc-statistical-reporting

5. In your Methods section, please provide additional information the demographic details of your participants, such as a table of relevant demographic details.

7. Thank you for stating the following financial disclosure:

'Drs. Kenji Fujiwara, Toshihiro Masatsugu, Atsushi Abe, Tatsuya Hirano, Masayuki Sada have no conflicts of interest or financial ties to disclose.'

8. Please include captions for your Supporting Information files at the end of your manuscript, and update any in-text citations to match accordingly. Please see our Supporting Information guidelines for more information: http://journals.plos.org/plosone/s/supporting-information

Reviewers' comments:

Reviewer's Responses to Questions

**Comments to the Author**

1. Is the manuscript technically sound, and do the data support the conclusions?

Reviewer #1: Partly

Reviewer #2: Yes

Reviewer #3: Partly

2. Has the statistical analysis been performed appropriately and rigorously? 

Reviewer #1: Yes

Reviewer #2: Yes

Reviewer #3: No

3. Have the authors made all data underlying the findings in their manuscript fully available?

Reviewer #1: No

Reviewer #2: Yes

Reviewer #3: Yes

4. Is the manuscript presented in an intelligible fashion and written in standard English?

Reviewer #1: Yes

Reviewer #2: Yes

Reviewer #3: No

5. Review Comments to the Author

Reviewer #1: 1. Is the manuscript technically sound, and do the data support the conclusions?

- the main limitation for this study has been stated by the authors themselves. The coding for the final diagnosis can be subjective and dependent on a single surgeon's opinion. It was not stated in the study how many cases had dual (or even triple) diagnoses. This could certainly be a huge confounder. Seeing as to the fact that the whole premise of the conclusion, including the survival analysis and the analysis done in Table 3 is based upon the diagnostic coding, this is the Achilles heel of the study. Perhaps if the number of cases with 2 or more diagnoses was reported (and if this number is low), it can be accepted that the confounder effect is minimal

- this study covers a period of over 28 years and differences in the treatment and prognosis of gallbladder cancer have not been taken into account in the survival analysis. For example, how many UGBC patients subsequently underwent completion radical cholecystectomy?

- another known prognostic factor for gallbladder cancer is whether there was unintentional gallbladder perforation with bile spillage intra-op, and whether the cancer was located on the hepatic or peritoneal part of the gallbladder. These variables were not reported in the study and could have an impact on the survival analysis.

2. Has the statistical analysis been performed appropriately and rigorously?

- Table 3 might be better represented by showing the odds ratio of age and thickened gallbladder wall specific to each diagnosis

3. Have the authors made all data underlying the findings in their manuscript fully available?

- the validation data that the authors performed (as mentioned in the Discussion) is not shown in detail

4. Is the manuscript presented in an intelligible fashion and written in standard English?

- Yes, although the sentence phrasing in some instances can be improved upon. Just to bring up one example, this sentence phrasing can be improved: 'We wanted to calculate the identification rates from large population, so we kept to use the diagnostic terms in our database'

Reviewer #2: The authors have undertaken a worthwhile study looking at unexpected gallbladder cancer (UGBC) in patients undergoing laparoscopic cholecystectomy. To strengthen the paper, a number of comments are made as follows:

Abstract

1) the Methods section needs more information added to inform the reader how the study was conducted

2) the Results sections needs to better align with the primary aim of the study and the detection rate of UGBC should be clearly stated (63 cases not 77).

Introduction

1) the aims of the study needs to be revisited as much of the results focuses on the pathological findings / prognosis / survival of UGBC and this is not reflected in the aims as they are currently stated

Methods

1) Greater detail needs to be added into how the statistical analysis was undertaken

Results

1) Suggest adding a sub-heading 'patient characteristics' to clearly describe this information rather than mixing it in with the other results

2) The detailed data outlined in Table 1 does not specifically relate to the aims of this study and there is also duplication with the data also being described in the text.

3) The description of how the data was analysed under sub-heading 'Indentification rates of UGBC by categorised with preoperative diagnoses' need to be moved to the Methods section and again there is duplication of both describing the data in written format and in Table 2.

4) The main study aim is to describe the overall UGBC rate and this is not clear in Table 2 or clearly described. It should be the first sentence.

5) How does the data described under sub-heading 'Analysis of pathological findings and prognosis of UGB' relate to the aims of this study? Either it needs to be added as a secondary aim or this is not relevant.

Discussion

1) Please clarify the UGBC rate as 0.7% based on 63 patients and not 77? (14 patients removed as had preop diagnosis of cancer?)

2) What are the clinical implications of these findings? Should practice be changed in any way? These should be discussed in some detail to make the findings more meaningful.

3) A clear limitation of this study is the small UGBC case numbers when analysed by preop diagnosis (as outlined in Table 3). This should be acknowledged and discussed.

4) The description regarding re-evaluation of 1615 patients with Tokyo guidelines should not be discussed without that data having been previously described in the results section. It either needs to be added to the results or removed.

Reviewer #3: Dear Dr Fujiwara, thank you for submitting the manuscript to PLOS ONE for review. I congratulate the team for keeping such good records for the last 28 years and analysing them.

Some comments that I have that will need to be reviewed by your team:

1. Please include the IRB number in the initial declaration as per Journal guidelines.

2. In the methods section, you mentioned that if more than one diagnostic term is recorded for a patient, the first term will be given the priority. I have concerns that this may introduce errors in your prediagnosis classification as patients with a more significant 'second' diagnosis may be missed. For example as you mention if a patient is classified as both 'Cholecystitis' and 'Polyp', they will be categorised as 'Cholecystitis' for your study, but the significance of having a polyp is much higher in terms of the risk of malignancy. It may be more suitable to classify patients according to their diagnosis which may carry a higher risk of malignancy rather than whichever came first.

3. Please include more details on which types of lesions are classified as benign tumors in your database

4. Please clarify regarding your data on 'Thickened Wall' in the results table, is this based on pre-operative imaging or -post-operative pathological finding?

5. There is an error in the percentage for benign disease 60 year old in Line 178

6. In the results and analysis, your have included the 14 patients with suspected Gall bladder Cancer in some tables but have omitted them in others. My concern is that the main aim of this paper is to investigate the incidence of unexpected Gall bladder Cancer in cholecystectomy patients. A pre-operative diagnosis of suspected Gall bladder Cancer would not fit into this definition, and including the 14 patients would not be suitable for your conclusion, as the number of patients with UGBC will actually be 63 and not 77.

7. In the discussion section, will you be able to comment on how this data will guide your clinical management of patients going for cholecystectomy?

Thank you.

6. PLOS authors have the option to publish the peer review history of their article (what does this mean?). If published, this will include your full peer review and any attached files.

Reviewer #1: No

Reviewer #2: No

Reviewer #3: No

---

## [Author Response · Author response to Decision Letter 0]

23 Aug 2020

Dear Editors and Reviewers,

We are very pleased to submit our revised manuscript, entitled “Preoperative Diagnoses and Identification Rates of Unexpected Gallbladder Cancer ”, by Fujiwara et al. for your consideration for publication in PLOS ONE.

Thank you for taking the time to review our manuscript. We appreciate your insightful comments and recommendations. We are grateful for the opportunity to revise our manuscript. Below, we have listed each comment, along with our subsequent revisions and responses. Also, we showed major changes with highlighted color as yellow in the file of “Revised manuscript with tracked change”. We kept any small change shown as the tracking function of MS because we had a lot of minor changes in English after English proofreading.

The changes in figures are described below:

New Figures or Panels:

S1 Table

S2 Table

S7 Table

Changes of the numbering in the figures:

S1 Table-> S3 Table

S2 Table -> S4 Table

S3 Table -> S5 Table

S4 Table -> S6 Table 

Editor:

Thank you for submitting your manuscript to PLOS ONE. After careful consideration, we feel that it has merit but does not fully meet PLOS ONE’s publication criteria as it currently stands. Therefore, we invite you to submit a revised version of the manuscript that addresses the points raised during the review process.

Author Response: We appreciate the issues to improve our manuscript raised by the editor and the reviewers. We have addressed these concerns by revising the manuscript and described the comments for these specific suggestions. 

(a) Perhaps, it would be good to try to shorten the initial part of the paper from Introduction to Results

Author Response: We appreciate the editor’s advice. We removed some sentences from the Introduction in order to shorten it. Some sentences were also included in Discussion, so we thought this was redundant as the editor pointed out. We removed these sentences from the Introduction, “For acute cholecystitis, we found two reports of massive surveys of acute cholecystitis; Thorbjarnarson reported the rate as 1.0% in 1960 and Kim et al. reported it as 1.6% in 2010. The UGBC identification rate in acute cholecystitis increased according to these reports. This was surprising because preoperative imaging analysis has dramatically improved in the last half-century, and current imaging can show more detailed images than what was possible in the past”.

(b) For the discussion section, I think it would be best for the authors to highlight what are the implications of the findings of the paper and the subsequent research questions that should be answered going ahead.

Author Response: Thank you for a great suggestion to improve the discussion section. We acknowledged that our previous statement was weak to show the finding of the paper, so we added the sentence, “Our UGBC identification rate was low compared with the rates in these studies. We suspect that the type of gallbladder disease in each facility could affect this rate. In our LSC cases, cholecystolithiasis was dominant (60.7%), and we found that UGBC was rarely found in cholecystolithiasis cases (0.054%). Kim et al. reported a percentage of acute cholecystitis in their LSC cases of 43.3%, which was much higher than ours (10.3%), and their incidental pathological UGBC detection rate was 1.0%. Differences in gallbladder diseases could have affected the differences in UGBC rates between the studies.” In addition, we added the sentence at the end of the second paragraph of the discussion section in order to show the possible subsequent research question, “We believe that even high-definition imaging cannot sufficiently distinguish the thickened walls of gallbladder cancers from those of acute cholecystitis preoperatively, but further evaluation is needed to determine the reason for the increased rate.”

(c) Please ensure that your manuscript meets PLOS ONE's style requirements, including those for file naming. The PLOS ONE style templates can be found at https://journals.plos.org/plosone/s/file?id=wjVg/PLOSOne_formatting_sample_main_body.pdf and https://journals.plos.org/plosone/s/file?id=ba62/PLOSOne_formatting_sample_title_authors_affiliations.pdf

Author Response: Thank you for informing us about this very important point. We changed the font-sizes and file-naming and revised the title page by following the instruction.

(d) Thank you for including your ethics statement: 

'Our institutional review board approved the use of this database for research purposes and waived the requirement for informed consent.' 

Please amend your current ethics statement to include the full name of the ethics committee/institutional review board(s) that approved your specific study. Once you have amended this/these statement(s) in the Methods section of the manuscript, please add the same text to the “Ethics Statement” field of the submission form (via “Edit Submission”).

Author Response: Thank you for confirming the important points. We added the information into the Method section, “Sada Hospital has its own Institutional Review Board (IRB) reviewing all the studies in the hospital. The IRB approved the use of the database for research purposes and waived the requirement for informed consent.” Also, we added the same text to the field of the submission form.

(e) In your ethics statement in the manuscript and in the online submission form, please provide additional information about the patient records used in your retrospective study.

Specifically, please ensure that you have discussed whether all data were fully anonymized before you accessed them.

Author Response: Thank you for the confirmation. We added the sentence into the ethics statement, “We fully anonymized all data before accessing the data.”

(f) To comply with PLOS ONE submission guidelines, in your Methods section, please provide additional information regarding your statistical analyses.

For more information on PLOS ONE's expectations for statistical reporting, please see https://journals.plos.org/plosone/s/submission-guidelines.#loc-statistical-reporting

Author Response: Thank you for pointing out about this important thing. We added the version of analyzing software like “GraphPad Prism Version 8.3.1”. We added the version of analyzing software like “GraphPad Prism Version 8.3.1”. Also, we changed the p-value by following the instruction. We showed the precise p-values for the larger than 0.001 and changed the lower than 0.001 p-values to p0.001.

(g) In your Methods section, please provide additional information the demographic details of your participants, such as a table of relevant demographic details.

Author Response: Thank you for your great advice. We provided demographic details as S1 Table.

(h) We note that you have included the phrase “data not shown” in your manuscript. Unfortunately, this does not meet our data sharing requirements. PLOS does not permit references to inaccessible data. We require that authors provide all relevant data within the paper, Supporting Information files, or in an acceptable, public repository. Please add a citation to support this phrase or upload the data that corresponds with these findings to a stable repository (such as Figshare or Dryad) and provide and URLs, DOIs, or accession numbers that may be used to access these data. Or, if the data are not a core part of the research being presented in your study, we ask that you remove the phrase that refers to these data.

Author Response: Thank you for the great advice. We added the S7 Table newly in order to show the undescribed data. For the other “data not shown” sentence, we deleted the sentences because we thought this is not a core part of the research. We changed the paragraph in the discussion, “We acknowledge that this study has limitations regarding the accuracy of the diagnostic terms. We used the diagnostic terms in the records reported by each surgeon, but the recorded diagnostic terms were not reassessed by other surgeons. Revalidation of all patients’ data was difficult because of the loss of records owing to storage expiration dates during the study period. A second limitation was the insufficient numbers of UGBC cases for evaluating the risk factors. We aimed to analyze data from as large a base as possible, so we decided to use the entire LSC database despite the difficulty confirming patients’ diagnoses. We kept all records for cancer cases during the 28-year period, and we are confident in the preoperative diagnoses for these cases. The entire database has been managed and supervised by one specific surgeon continuously for 28 years; therefore, we believe that the criteria remained the same for the diagnoses.”

(i) Thank you for stating the following financial disclosure:

'Drs. Kenji Fujiwara, Toshihiro Masatsugu, Atsushi Abe, Tatsuya Hirano, Masayuki Sada have no conflicts of interest or financial ties to disclose.'

a. Please clarify the sources of funding (financial or material support) for your study. List the grants or organizations that supported your study, including funding received from your institution.

d. If you did not receive any funding for this study, please state: “The authors received no specific funding for this work.”

Author Response: Thank you for asking about this important point. We added the sentence into the part of the conflict of the interest, “The authors received no specific funding for this work.”

(j) Please include captions for your Supporting Information files at the end of your manuscript, and update any in-text citations to match accordingly. Please see our Supporting Information guidelines for more information: http://journals.plos.org/plosone/s/supporting-information

Author Response: Thank you for informing us. We added captions for our Supporting Information files at the end of our manuscript, and also updated the in-text citations.

 

Reviewer #1:

1-1. Is the manuscript technically sound, and do the data support the conclusions?

- the main limitation for this study has been stated by the authors themselves. The coding for the final diagnosis can be subjective and dependent on a single surgeon's opinion. It was not stated in the study how many cases had dual (or even triple) diagnoses. This could certainly be a huge confounder. Seeing as to the fact that the whole premise of the conclusion, including the survival analysis and the analysis done in Table 3 is based upon the diagnostic coding, this is the Achilles heel of the study. Perhaps if the number of cases with 2 or more diagnoses was reported (and if this number is low), it can be accepted that the confounder effect is minimal

Author Response: Thank you for asking very important points. 

1) We agree that the diagnoses decided by one surgeon after surgery could be subjective. However, we think that the good point of the study is this study was managed and supervised for 28 years by one specific surgeon (MS) continuously since 1991. Therefore, we think that we almost kept the same criteria since the starting point. Once, we thought to restrict the data only for 10 years in order to confirm all data again, but it will cause a great decrease in the total number, and also the cancer cases would be one-third. Fortunately, we have kept all information on cancer cases so that we could confirm the data of the cancer cases. We utilized the whole data in order to calculate the identification rate from the data as large as possible. We arranged the discussion part, “We acknowledge that this study has limitations regarding the accuracy of the diagnostic terms. We used the diagnostic terms in the records reported by each surgeon, but the recorded diagnostic terms were not reassessed by other surgeons. Revalidation of all patients’ data was difficult because of the loss of records owing to storage expiration dates during the study period. A second limitation was the insufficient numbers of UGBC cases for evaluating the risk factors. We aimed to analyze data from as large a base as possible, so we decided to use the entire LSC database despite the difficulty confirming patients’ diagnoses. We kept all records for cancer cases during the 28-year period, and we are confident in the preoperative diagnoses for these cases. The entire database has been managed and supervised by one specific surgeon continuously for 28 years; therefore, we believe that the criteria remained the same for the diagnoses.”

2) We agree that our study was not clear about how to select the preoperative diagnoses from two or three diagnostic terms. This is also being pointing out by other reviewers, and we changed these criteria, dramatically. We summarized new criteria in the Method section, “We found 1239 cases (13.5% in 9200 cases) with two or three diagnostic terms simultaneously, and we summarized these in S2 Table. We prioritized the main diagnostic terms, such as cholecystitis, when patients had two diagnostic terms, such as gallbladder polyp, coincidentally found in the gallbladder with cholecystitis. When patients had multiple diagnostic terms, including choledocholithiasis, we prioritized other diagnostic terms because gallbladder lesions are usually related to the risk of malignancy more than with choledocholithiasis. In cases with concurrent cholecystolithiasis and gallbladder polyp, we chose gallbladder polyp (as a benign tumor) to categorize the disease according to the preoperative diagnosis because gallbladder polyp may carry a higher risk of malignancy.” As we wrote in the above-cited sentences, we summarized the details of the information of the patients having more than two diagnostic terms in the S2 table newly made. We had 1239 cases, but most of the cases (971 cases) were with adenomyomatosis found coincidentally or with choledocholithiasis. We think the remaining will not affect the data coding strongly.

1-2. - this study covers a period of over 28 years and differences in the treatment and prognosis of gallbladder cancer have not been taken into account in the survival analysis. For example, how many UGBC patients subsequently underwent completion radical cholecystectomy?

Author Response: Thank you for asking about this important point. Not so big change happened for 28 years about the treatment, but the improvement of imaging might have increased the detection rate of the more malignant-suspicious lesions. 

We previously analyzed the data about re-exploration surgeries and adjuvant therapies. Twenty-two of 46 cancer cases for which data existed underwent conversion to laparotomy or secondary surgery, such as hepatectomy. For eight other cases, laparoscopic whole layer cholecystectomies were performed. The other 16 cases did not undergo laparotomy, secondary surgery, or whole layer cholecystectomy. The 5 cases in them were Tis/T1. The other 11 patients in them (all of which were T2) rejected the procedure and the reasons of the rejection were the advanced age of the patients or the existence of lymph or liver metastases. The patients not undergoing additional surgery, except for two cases with distant metastases, nine patients survived throughout the observation period. In our facility, adjuvant chemotherapy with gemcitabine or tegafur/gimeracil/oteracil was done when lymph node metastases were found. We did not find any relationship between the prognosis and the type of surgery or adjuvant chemotherapy. 

Firstly, we added this information into the manuscript, but the manuscript became too long. We thought this information could be redundant because our purpose is to detect the identification rate. Therefore, we omitted this information in this manuscript.

1-3. - another known prognostic factor for gallbladder cancer is whether there was unintentional gallbladder perforation with bile spillage intra-op, and whether the cancer was located on the hepatic or peritoneal part of the gallbladder. These variables were not reported in the study and could have an impact on the survival analysis.

Author Response: Thank you for asking about important points. We reanalyzed with the bile spillage of intraoperative perforation, the cancer location, and the prognosis. We did not find any relationship among them. In 77 cancer cases, 62 cases had the record of the precise information about intraoperative perforation. Forty- three patients did not have the perforation and 19 patients had the perforation. We analyzed the relationship with the perforation, the preoperative diagnoses, and the prognosis and we did not find any tendency. Also, we analyzed the location of the tumor (liver side; 8 cases, peritoneum side; 7 cases, spreading to the whole layer of the gallbladder or existing on both sides; 41cases) and we did not find any tendency about the prognosis with them.

2. Has the statistical analysis been performed appropriately and rigorously?

- Table 3 might be better represented by showing the odds ratio of age and thickened gallbladder wall specific to each diagnosis.

Author Response: Thank you for pointing out an important point. We tried to add the odds ration into Table 3, but data would be bigger and busy. We decide not to add the odds ratio.

3. Have the authors made all data underlying the findings in their manuscript fully available?

- the validation data that the authors performed (as mentioned in the Discussion) is not shown in detail

Author Response: Thank you for detecting very important points. The editor and the other reviewers also pointed out this point. We had two “data not shown” sentences in the discussion, so we added one supplementary table, S7 Table, “Identification rates of incidentally pathological detected gallbladder cancer cases categorized with preoperative diagnoses.”. On the other hand, we deleted the other sentence including “data not shown” about the validation of the preoperative diagnoses. We changed the sentence as like we explained in question 1-1.

4. Is the manuscript presented in an intelligible fashion and written in standard English?

- Yes, although the sentence phrasing in some instances can be improved upon. Just to bring up one example, this sentence phrasing can be improved: 'We wanted to calculate the identification rates from large population, so we kept to use the diagnostic terms in our database'

Author Response: Thank you for detecting very important points. We submitted English proofreading in order to improve the quality of English.

 

Reviewer #2:

The authors have undertaken a worthwhile study looking at unexpected gallbladder cancer (UGBC) in patients undergoing laparoscopic cholecystectomy. To strengthen the paper, a number of comments are made as follows:

Abstract

1) the Methods section needs more information added to inform the reader how the study was conducted.

Author Response: Thank you for advising about improving our manuscript. Unfortunately, in the abstract, we shortened the method section due to the limitation of the numbers of the words in the abstract. We considered adding more information there, but we thought the readers could understand how to proceed with the study from the whole of the abstract. Therefore, we did not long the method section.

2) the Results sections needs to better align with the primary aim of the study and the detection rate of UGBC should be clearly stated (63 cases not 77).

Author Response: Thank you for detecting very important points. We reconstructed the result part and also changed the UGBC identification rate by decreasing the total numbers from 77 to 63. We changed like this, “Results: The UGBC identification rate was 0.69% (63/9186 patients). The UGBC identification rates categorized according to the preoperative diagnoses were 1.3% (13/969) for acute cholecystitis, 2.4% (16/655) for benign tumor, 2.0% (28/1383) for chronic cholecystitis or cholecystitis, and 0.054% (3/5585) for cholecystolithiasis.”

Introduction

1) the aims of the study needs to be revisited as much of the results focuses on the pathological findings / prognosis / survival of UGBC and this is not reflected in the aims as they are currently stated.

Author Response: Thank you for advising about improving our manuscript. We revisited the introduction, and arranged this, “Löhe et al. reported that only 50% of gallbladder cancers are recognized preoperatively. There could be several reasons for the difficulty recognizing cancers preoperatively. For example, the thickened tumor wall is sometimes difficult to distinguish from wall thickening owing to inflammation in cholecystitis. Additionally, early-stage cancers and/or flat-type cancers are difficult to diagnose. Several reports showed the risk factors for malignancy of gallbladder lesions. In this study, we evaluated the efficacy of using the reported risk factors to predict the UGBC and to determine the factors suspicious for cancer, especially when categorizing the risk factors according to preoperative diagnoses. In addition, we compared the pathological findings and the prognosis of gallbladder cancer cases with preoperative diagnoses. Identifying differences in gallbladder cancer progression according to preoperative diagnoses might provide clues to determine poor prognostic factors.”

Methods

1) Greater detail needs to be added into how the statistical analysis was undertaken

Author Response: Thank you for asking about this point. We arranged the statistical section, “Patients’ characteristics and pathological findings were analyzed using the Chi-square test. Survival data for the Kaplan–Meir analysis was performed using the log-rank test. Statistical analysis and graphic presentations were performed using GraphPad Prism Version 8.3.1 (GraphPad Software, San Diego, CA, USA). A p-value of 0.05 was considered statistically significant.”

Results

1) Suggest adding a sub-heading 'patient characteristics' to clearly describe this information rather than mixing it in with the other results

Author Response: Thank you for asking about important point. We reconsidered this section and split the section into the method section and the Result section. In the Method section, we used the recommended subtitle “Patient characteristics “. 

2) The detailed data outlined in Table 1 does not specifically relate to the aims of this study and there is also duplication with the data also being described in the text.

Author Response: Thank you for detecting very important points. We acknowledge this part contained redundancy so we moved the patient characteristics to the method section. To show the percentage of gallbladder cancer in our LSC cases was not our aim, but we knew many manuscripts showed this percentage as the main percentage of cancer. Therefore, we showed this percentage in the first paragraph before starting to show the UGBC identification rate in order to avoid the confusion of the readers. We deleted the duplication from the paragraph, “The number of patients with cholecystitis or chronic cholecystitis was 1355 (14.7%), and the number of patients with a diagnosis of acute cholecystitis was 956 (10.3%).”. Thank you for noticing us.

3) The description of how the data was analysed under sub-heading 'Indentification rates of UGBC by categorised with preoperative diagnoses' need to be moved to the Methods section and again there is duplication of both describing the data in written format and in Table 2.

Author Response: Thank you for detecting very important points. We acknowledge this part should be in the method, and we already write the same thing in the method section. Therefore, we just removed, “-based on the patients’ symptoms, blood test results, imaging findings, and preoperative diagnosis”. Also, we shortened the sentences in order to avoid writing just duplication.

4) The main study aim is to describe the overall UGBC rate and this is not clear in Table 2 or clearly described. It should be the first sentence.

Author Response: Thank you for detecting very important points. We arranged the order of the sentences, “To analyze the data for gallbladder cancers according to the preoperative diagnostic terms, we categorized 77 gallbladder cancer cases according to the preoperative diagnoses and calculated the UGBC identification rates for each preoperative diagnosis (Table 2 and S3 Table). Of 9186 cases, after excluding 14 gallbladder cancer patients suspected preoperatively of having malignancy, the number of UGBC patients was 63, and the UGBC identification rate was 0.69%. The UGBC identification rate in patients with benign tumor was highest (2.4%), followed by chronic cholecystitis or cholecystitis (2.0%), and acute cholecystitis (1.3%). The UGBC identification rate in cases with cholecystolithiasis was lowest (0.054%).”

5) How does the data described under sub-heading 'Analysis of pathological findings and prognosis of UGBC' relate to the aims of this study? Either it needs to be added as a secondary aim or this is not relevant.

Author Response: Thank you for detecting very important points. We agreed that our aim lacked the reasons of the evaluation of pathological findings. We added the sentences into the introduction, “In addition, we compared the pathological findings and the prognosis of gallbladder cancer cases with preoperative diagnoses. Identifying differences in gallbladder cancer progression according to preoperative diagnoses might provide clues to determine poor prognostic factors.” 

Discussion

1) Please clarify the UGBC rate as 0.7% based on 63 patients and not 77? (14 patients removed as had preop diagnosis of cancer?)

Author Response: Thank you for detecting very important points. We changed the UGBC tare to 0.7% by changing the total number of UGBC to 63.

2) What are the clinical implications of these findings? Should practice be changed in any way? These should be discussed in some detail to make the findings more meaningful.

Author Response: Thank you for detecting very important points. We tried to improve our discussion part more, and add this point, “Our UGBC identification rate was low compared with the rates in these studies. We suspect that the type of gallbladder disease in each facility could affect this rate. In our LSC cases, cholecystolithiasis was dominant (60.7%), and we found that UGBC was rarely found in cholecystolithiasis cases (0.054%). Kim et al. reported a percentage of acute cholecystitis in their LSC cases of 43.3%, which was much higher than ours (10.3%), and their incidental pathological UGBC detection rate was 1.0%. Differences in gallbladder diseases could have affected the differences in UGBC rates between the studies.”To be honest, we wanted to describe preoperative diagnosis as the possibility of worse prognosis factors, but our total number of UGBC was still small and we needed to study with more huge numbers of cancer patients about the worse prognosis factor. In this study, we only described that the pre-diagnosis of acute cholecystitis might be the factor of more advanced cancer cases compared with the pre-diagnosis of chronic cholecystitis. We want to evaluate this point in the future.

3) A clear limitation of this study is the small UGBC case numbers when analysed by preop diagnosis (as outlined in Table 3). This should be acknowledged and discussed.

Author Response: Thank you for detecting very important points. We describe this point in the discussion, “We acknowledge that this study has limitations regarding the accuracy of the diagnostic terms. We used the diagnostic terms in the records reported by each surgeon, but the recorded diagnostic terms were not reassessed by other surgeons. Revalidation of all patients’ data was difficult because of the loss of records owing to storage expiration dates during the study period. A second limitation was the insufficient numbers of UGBC cases for evaluating the risk factors. We aimed to analyze data from as large a base as possible, so we decided to use the entire LSC database despite the difficulty confirming patients’ diagnoses. We kept all records for cancer cases during the 28-year period, and we are confident in the preoperative diagnoses for these cases. The entire database has been managed and supervised by one specific surgeon continuously for 28 years; therefore, we believe that the criteria remained the same for the diagnoses.”

4) The description regarding re-evaluation of 1615 patients with Tokyo guidelines should not be discussed without that data having been previously described in the results section. It either needs to be added to the results or removed.

Author Response: Thank you for detecting very important points. We agreed that this part is controversial and we thought this part could be only redundant. We removed the information and changed the sentence as like we explained the previous question. 

 

Reviewer #3: 

Dear Dr Fujiwara, thank you for submitting the manuscript to PLOS ONE for review. I congratulate the team for keeping such good records for the last 28 years and analysing them.

Some comments that I have that will need to be reviewed by your team:

1. Please include the IRB number in the initial declaration as per Journal guidelines.

Author Response: Thank you for asking about very important point. We showed the IRB number in the Method section, “The IRB approved the use of the database for research purposes and waived the requirement for informed consent (IRB number: S190726-1).”

2. In the methods section, you mentioned that if more than one diagnostic term is recorded for a patient, the first term will be given the priority. I have concerns that this may introduce errors in your prediagnosis classification as patients with a more significant 'second' diagnosis may be missed. For example as you mention if a patient is classified as both 'Cholecystitis' and 'Polyp', they will be categorised as 'Cholecystitis' for your study, but the significance of having a polyp is much higher in terms of the risk of malignancy. It may be more suitable to classify patients according to their diagnosis which may carry a higher risk of malignancy rather than whichever came first.

Author Response: Thank you for asking very important points. We agree that this was not clear about how to select the preoperative diagnoses from two or three diagnostic terms. This is also being pointing out by other reviewers, and we changed this criteria, dramatically. We summarized new criteria in the Method section, “We found 1239 cases (13.5% in 9200 cases) with two or three diagnostic terms simultaneously, and we summarized these in S2 Table. We prioritized the main diagnostic terms, such as cholecystitis, when patients had two diagnostic terms, such as gallbladder polyp, coincidentally found in the gallbladder with cholecystitis. When patients had multiple diagnostic terms, including choledocholithiasis, we prioritized other diagnostic terms because gallbladder lesions are usually related to the risk of malignancy more than with choledocholithiasis. In cases with concurrent cholecystolithiasis and gallbladder polyp, we chose gallbladder polyp (as a benign tumor) to categorize the disease according to the preoperative diagnosis because gallbladder polyp may carry a higher risk of malignancy.” As we wrote in the above-cited sentences, we summarized the details of the information of the patients having more than two diagnostic terms in the S2 table newly made. We had 1239 cases, but most of the cases (971 cases) were with adenomyomatosis found coincidentally or with choledocholithiasis. We think the remaining will not affect the data coding strongly.

3. Please include more details on which types of lesions are classified as benign tumors in your database

Author Response: Thank you for asking about very important points. We inserted these sentences into the method section, “Benign tumor” meant gallbladder protruded lesions, including gallbladder pseudo-polyps such as cholesterol polyp and also true polyps such as adenomas, pathologically diagnosed as benign. We categorized “adenomyomatosis” separately from “benign tumor” because typical adenomyomatosis lesions are distinguishable from gallbladder polyps by imaging.”

4. Please clarify regarding your data on 'Thickened Wall' in the results table, is this based on pre-operative imaging or -post-operative pathological finding?

Author Response: Thank you for pointing out this very important point. We noticed that this part could make the confusion for the readers. Thickened wall was evaluated after the surgery, so this is post-operative finding. We already wrote about this in the method section, “Wall thickness was determined postoperatively by evaluating the surgical specimen.” However, we acknowledge this is confusing point, so we emphasize this point in the manuscript by changing the description in the result, “Preoperative data included age, sex, and gallbladder imaging on drip infusion cholangiography with CT (DIC-CT), and gallbladder wall thickness was confirmed after surgery by evaluating the surgical specimen.”We also changed the discussion part, “Our data also showed that UGBCs diagnosed preoperatively as benign tumors had thickened walls more frequently than cases finally diagnosed as benign disease. Although we examined thickened walls postoperatively, other studies reported that a preoperative finding of thickened walls could also indicate UGBC. However, a thickened wall may not be a useful UGBC risk factor for other UGBC cases, such as UGBCs in acute cholecystitis. Resected specimens in acute cholecystitis usually have thickened walls (95.5% in our data).” 

5. There is an error in the percentage for benign disease >60 year old in Line 178

Author Response: Thank you for detecting the mistake. We revised the part, “The comparison between two tables (S4 and S5 Tables) showed that the percentage of older age (≥60 years old) was significantly higher in UGBCs (68.7%–100%) than cases finally diagnosed as benign (21.2%–47.8%) in each preoperative-diagnosis group (p=0.0014 or lower) except for UGBC patients pre-diagnosed with cholecystolithiasis (Table 3).”

6. In the results and analysis, your have included the 14 patients with suspected Gall bladder Cancer in some tables but have omitted them in others. My concern is that the main aim of this paper is to investigate the incidence of unexpected Gall bladder Cancer in cholecystectomy patients. A pre-operative diagnosis of suspected Gall bladder Cancer would not fit into this definition, and including the 14 patients would not be suitable for your conclusion, as the number of patients with UGBC will actually be 63 and not 77.

Author Response: Thank you for detecting very important points. We changed the UGBC rate to 0.7% by changing the total number of UGBC to 63.

7. In the discussion section, will you be able to comment on how this data will guide your clinical management of patients going for cholecystectomy?

Author Response: Thank you for asking about very important points. Unfortunately, we think that our data did not have the impact to change the management of patients going for cholecystectomy. However, the surgeons could be more cautious for the surgery of cholecystitis compared to the surgery of cholecystolithiasis due to higher UGBC identification rates. For example, the surgeon may dissect lymph nodes as much as possible for the cholecystitis surgery in case of UGBC. However, we thought there was big leap of the theory, so we did not add our idea about this into the discussion.

---

## [Editor Report · Decision Letter 1]

2 Sep 2020

Preoperative Diagnoses and Identification Rates of Unexpected Gallbladder Cancer

PONE-D-20-15344R1

Dear Dr. Fujiwara,

We’re pleased to inform you that your manuscript has been judged scientifically suitable for publication and will be formally accepted for publication once it meets all outstanding technical requirements.

Kind regards,

Ker-Kan Tan

Academic Editor

PLOS ONE
---

## [Editor Report · Acceptance letter]

4 Sep 2020

PONE-D-20-15344R1 

Preoperative diagnoses and identification rates of unexpected gallbladder cancer 

Dear Dr. Fujiwara:

I'm pleased to inform you that your manuscript has been deemed suitable for publication in PLOS ONE. Congratulations! Your manuscript is now with our production department. 

Kind regards, 

on behalf of

Dr. Ker-Kan Tan 

Academic Editor

PLOS ONE